# Towards Understanding the Key Signature Pathways Associated from Differentially Expressed Gene Analysis in an Indian Prostate Cancer Cohort

**DOI:** 10.3390/diseases11020072

**Published:** 2023-05-11

**Authors:** Nidhi Shukla, Bhumandeep Kour, Devendra Sharma, Maneesh Vijayvargiya, T. C. Sadasukhi, Krishna Mohan Medicherla, Babita Malik, Bhawana Bissa, Sugunakar Vuree, Nirmal Kumar Lohiya, Prashanth Suravajhala

**Affiliations:** 1Department of Biotechnology and Bioinformatics, Birla Institute of Scientific Research (BISR), Statue Circle, Jaipur 302001, India; 2Department of Chemistry, School of Basic Sciences, Manipal University Jaipur, Jaipur 303007, India; 3Department of Biotechnology, School of Bioengineering and Biosciences, Lovely Professional University, Phagwara 144001, India; 4Department of Urology, Rukmani Birla Hospital, Jaipur 302018, India; 5Department of Pathology, Mahatma Gandhi University of Medical Sciences and Technology, Jaipur 302022, India; 6Department of Urology, Mahatma Gandhi University of Medical Sciences and Technology, Jaipur 302022, India; 7Department of Bioengineering, Birla Institute of Technology, Mesra Jaipur Campus, 27-Malaviya Industrial Area, Jaipur 302017, India; 8Department of Biochemistry, Central University of Rajasthan, Ajmer 305817, India; 9Bioclues.org, Hyderabad 500072, India; 10MNR Foundation for Research & Innovation, MNR University, Sangareddy 502294, India; 11Department of Zoology, Center for Advanced Studies, University of Rajasthan, Jaipur 302004, India; 12Amrita School of Biotechnology, Amrita Vishwa Vidyapeetham, Kollam 690525, India

**Keywords:** prostate cancer, RNA-sequencing, differentially expressed genes, long non-coding RNAs

## Abstract

Prostate cancer (PCa) is one of the most prevalent cancers among men in India. Although studies on PCa have dealt with genetics, genomics, and the environmental influence in the causality of PCa, not many studies employing the Next Generation Sequencing (NGS) approaches of PCa have been carried out. In our previous study, we identified some causal genes and mutations specific to Indian PCa using Whole Exome Sequencing (WES). In the recent past, with the help of different cancer consortiums such as The Cancer Genome Atlas (TCGA) and International Cancer Genome Consortium (ICGC), along with differentially expressed genes (DEGs), many cancer-associated novel non-coding RNAs have been identified as biomarkers. In this work, we attempt to identify differentially expressed genes (DEGs) including long non-coding RNAs (lncRNAs) associated with signature pathways from an Indian PCa cohort using the RNA-sequencing (RNA-seq) approach. From a cohort of 60, we screened six patients who underwent prostatectomy; we performed whole transcriptome shotgun sequencing (WTSS)/RNA-sequencing to decipher the DEGs. We further normalized the read counts using fragments per kilobase of transcript per million mapped reads (FPKM) and analyzed the DEGs using a cohort of downstream regulatory tools, viz., GeneMANIA, Stringdb, Cytoscape-Cytohubba, and cbioportal, to map the inherent signatures associated with PCa. By comparing the RNA-seq data obtained from the pairs of normal and PCa tissue samples using our benchmarked in-house cuffdiff pipeline, we observed some important genes specific to PCa, such as STEAP2, APP, PMEPA1, PABPC1, NFE2L2, and HN1L, and some other important genes known to be involved in different cancer pathways, such as COL6A1, DOK5, STX6, BCAS1, BACE1, BACE2, LMOD1, SNX9, CTNND1, etc. We also identified a few novel lncRNAs such as LINC01440, SOX2OT, ENSG00000232855, ENSG00000287903, and ENST00000647843.1 that need to be characterized further. In comparison with publicly available datasets, we have identified characteristic DEGs and novel lncRNAs implicated in signature PCa pathways in an Indian PCa cohort which perhaps have not been reported. This has set a precedent for us to validate candidates further experimentally, and we firmly believe this will pave a way toward the discovery of biomarkers and the development of novel therapies.

## 1. Introduction

Prostate cancer (PCa) is the fifth most common cause of cancer death in males worldwide and the second most frequently diagnosed malignancy in many developed nations. In the recent past, there is an increased occurrence of PCa varying among different races and countries, particularly between Western (Caucasian) and Eastern (Asian) populations [1,2]. Notwithstanding the increased burden of PCa incidence in affluent nations, males of African ancestry (MAA) bear a disproportionate amount of this burden. Men living in the Caribbean and in Sub-Saharan Africa had the highest PCa mortality rates in the world. According to the International Agency for Research on Cancer (IARC), PCa is also anticipated to trend higher across Africa, where fatalities from the disease will rise from about 28,000 in 2010 to just over 57,000 by 2030.

Over the last few years, there has been an increase in PCa cases across all the different parts of and regions in India [3,4]. Next Generation Sequencing (NGS) has advanced our understanding of the diseased phenotypes, particularly cancers with tumor heterogeneity, and capture of large amounts of genetic architecture leading to isoforms, splice junctions, and post-transcriptional modifications with great accuracy and sensitivity [5,6]. Although studies have dealt with the genetics, genomics, and environmental influence in the causality of PCa, no association of genotype and phenotype employing the NGS approaches has been discussed in the Indian population. Recent studies from our lab using the whole exome sequencing (WES) approach of PCa have yielded very promising results to identify causal genes specific to PCa, with 30 causal genes and over eight genes specific to the Indian phenotype with variable degrees of genetic disposition [7]. Whole transcriptome shotgun sequencing (WTSS) or RNA-sequencing (RNA-seq), on the other hand, is used to study gene expression patterns, regulation, developmental biology, and clinical and health sectors [8]. The expected outcome of RNA-seq is the identification of differentially expressed genes (DEGs) in different conditions (for, e.g., wild type vs. mutant or control vs. tumor), which could provide a detailed mechanism of underlying disease [9]. The DEGs are selected based on the cut-off and the log2 fold change that depends on *p*-values obtained by statistical modeling [10]. RNA-seq generates millions of short reads that are aligned to a reference genome using different alignment software, and based on this, the characteristic of a particular dataset is calculated [11]. Although RNAs are labile and unstable in alkaline conditions, they can be easily detected and quantified at very low abundance for varying gene expression patterns; which, makes them suitable to use as biomarkers [12]. Compared with DNA and protein biomarkers, RNA biomarkers have more sensitivity and specificity and are very cost-effective [13]. Similarly, RNA biomarkers have the advantage of providing dynamic insights into cellular states and regulatory processes compared to DNA biomarkers [14]. With the advent of high-throughput sequencing technologies, different types of non-coding RNAs (e.g., small nuclear RNA, micro-RNA, small nucleolar RNA, long non-coding RNA, etc.) and protein-coding RNAs (i.e., mRNAs) have been detected [15]. Interestingly, there have been lots of novel non-coding RNAs discovered recently, out of which mRNAs, piwiRNAs, siRNAs, ceRNAs, and miRNAs have been well-documented as diagnostic and prognostic markers in different types of cancers (ovary, lung, breast, colorectal) [16]. In addition to small RNAs, long non-coding RNAs (lncRNAs) previously considered as ‘transcriptional noises’ have been known to have a diverse and significant role in various diseases, primarily cancers [17,18]. Although well-known lncRNAs in the form of MALAT, HOTAIR, XIST have been reported in genitourinary cancers, the increasing number of cases of PCa worldwide and as well as in the Indian population, warrants the need for identifying lncRNA-based biomarkers [19,20]. Further with the advent of RNA-seq, the identification of lncRNAs will hopefully facilitate the translational research to the bench side [21]. Therefore, with considerable data available on the Indian population, transcriptome analysis on Indian patients supplementing strategies in diseased phenotypes of PCa will be of great interest [22]. In the current study, we analyzed the RNA-seq data from a cohort of six patients and identified DEGs as well as lncRNAs which could lay emphasis as deterministic markers. Cufflinks-Cuffdiff has been used for transcriptome analysis in a wide number of studies, e.g., Tripathi et al. [23] have identified DEGs from breast cancer. Similarly, Kim et al. [24] have done transcriptome analysis of sinensetin-treated liver cancer cells using the Cufflinks-Cuffdiff pipeline. There are many more studies that corroborate the importance of these tools. We discuss the impending results obtained from this study and attempt to further compare and validate from our downstream analysis.

## 2. Materials and Methods

### 2.1. Patients, Clinical Samples, and Criteria

From a cohort of 60, we screened 6 patients who underwent prostatectomy (4 cases and 2 controls) from Rukmani Birla Hospital (RBH) (Table 1). The study was carried out through our CA Prostate Consortium of India (CAPCI; https://bioclues.org/capci last accessed on 13 April 2023) and received approval from the Institutional Ethics Committee (IEC) of Rukmani Birla Hospitals, Jaipur, India, and informed consent was judiciously taken. The inclusion and exclusion criteria for selecting patients have been mentioned in Table 2.

### 2.2. Tissue Preparation and RNA Sequencing

RNA was isolated using the RNeasy FFPE kit (Qiagen, Catalog No-73504) from BPH and malignant Formalin-Fixed Paraffin-Embedded (FFPE) blocks, and was sent for sequencing (outsourced). From approximately 0.5 mg of cross-section, the RNA was prepared wherein the NEBNext^®^ Ultra™ II Directional RNA Library Prep Kit was used for preparing the libraries following the manufacturer’s protocol. While 100 ng of FFPE RNA was used as input, it was then subjected to end repair and Illumina-specific adaptors were ligated. The adaptor-ligated product was then barcoded and subjected to 15 cycles of PCR. The samples after PCR were cleaned up using AMPure XP beads with the final libraries checked for quality using Qubit Fluorometer and Agilent Tapestation. The obtained libraries were pooled and diluted to the final optimal loading concentration before cluster amplification on the Illumina flow cell. Once the cluster generation is completed, the cluster flow cell is loaded on an Illumina HiSeq X instrument to generate 60 M, 150 bp paired end reads.

### 2.3. Bioinformatics and Downstream RNA-Sequencing Analysis

The Cufflinks-Cuffdiff pipeline was employed to yield significant changes at the level of transcript expression [25], as we used our benchmarked pipeline from our lab to run through the workflow [26]. The sequences were aligned to human genome reference (build hg38) using HISAT2 to produce the alignment results output in SAM (sequence alignment map/file). HISAT2 aligns a set of unpaired reads (in fastq or fq format) to the reference genome using the Ferragina and Manzini (FM)-index [27]. Cufflinks uses this map (SAM) and assembles the reads into transcripts, estimates their abundances, and finally examines DEGs from the samples (Figure 1). This was followed by Cuffdiff to check the DEGs which compared the aligned reads from RNA-seq samples from two or more conditions, and identified transcripts that are differentially expressed using a rigorous FPKM normalization/statistical analysis [28]. In the current study, Cuffdiff was used to perform differential analysis between the control samples and the other 4 malignant samples, respectively [29]. A *p*-value cutoff of 0.05 and less was used to identify the significantly expressed transcripts (Figure 2). We performed a real-time PCR validation for some selected DEGs. The primers were ordered accordingly, and RT-PCR was performed (Appendix A). We saw a significant difference in the expression of a few genes in malignant samples compared to control ones. But some of the genes did not show any difference which could be due to the small sample size as well as the poor quality of FFPE blocks.

### 2.4. Interaction Networks, Statistical Analysis, Gene Ontology, and Cbioportal Analyses

We generated an interaction network considering a flexible and intuitive approach to evaluating gene lists for generating l functional studies [30]. To check this, GeneMANIA (https://genemania.org/ last accessed on 13 April 2023) was employed which is a versatile, user-friendly web tool for developing gene function hypotheses, reviewing gene lists, and ranking genes for functional experiments [31]. In addition, we also used the search tool for the retrieval of interacting genes (STRING) database (https://string-db.org/ last accessed on 13 April 2023), and integrated DEGs into STRING to evaluate the interaction. Experimentally valid interactions with a score of a minimum of 0.4 (40% or more) were chosen to be ideal ones with the resulting file saved as a tab-separated values (TSV) file [32]. The raw data files from the STRING database were then imported into Cytoscape 3.5.1 and the cytoHubba (http://hub.iis.sinica.edu.tw/cytohubba/ last accessed on 13 April 2023) plugin was employed with clustering coefficient, betweenness, and closeness centralities to calculate the significant modules in the PPI network [33]. Different Cytoscape plugins can score and rank the nodes using different algorithms [34]. CytoHubba is one such plugin that uses a simple interface to analyze the different networks. CytoHubba implements eleven nodes such as degree, Maximum Neighborhood Component, betweenness, closeness, clustering coefficient, stress, etc., to rank any network [35]. Each method has an F function attached to it that gives each node v a numerical value (v). If a node’s score, or F (u), is higher than that of another node, or F(v), then we can say that node’s ranking is higher than that of that other node. PANTHER (http://www.pantherdb.org/ last accessed on 13 April 2023) is a gene ontology-based functional annotation tool that takes a variety of inputs such as Gene IDs, UniProtKB IDs, Ensembl IDs, etc., and results in either functional analysis or statistical enrichment analysis [36]. Furthermore, we deemed to check the expression and mutational profile of some of the genes from our study in the cbioportal (https://www.cbioportal.orglast last accessed on 13 April 2023) for cancer genomics that provides visualization, analysis, and the download of large-scale cancer genomics data sets [37]. We used data from TCGA, PanCancer Atlas of Prostate Adenocarcinoma where 489 samples/patients were screened [38].

## 3. Results

### 3.1. Distinct DEGs Were Obtained

Although there has been difficulty in extracting RNA from FFPE blocks from the RNA-seq, which is usually the case [39], while checking it and downstream analyses, all samples yielded good quality reads from FastQC with ca. 40 M transcript reads with no exposure in tiles. With the ensuing Cufflinks-Cuffdiff pipeline, we obtained approximately 70 DEGs, among which 65 were upregulated and 5 were downregulated with inherent *p*-value heuristics ≤ 0.05 and ≤−2 Log2FC and Log2FC ≥ 2 (Table 3).

Among the top niche-specific DEGs, collagen type VI α1 chain (COL6A1), a gene which is located on chromosome 21, encoding the α1 (VI) chain of type VI collagen (which is a primary extracellular matrix protein) was found which maintains the integrity of various tissues [40]. The signaling role of COL6A1 is very important in tumors as it increases tumor cell proliferation in osteosarcoma as well as promotes vascular invasion and distant metastasis in pancreatic carcinoma [41,42]. Also, COL6A1 plays a role as an oncogene in breast and lung cancer where it regulates anti-apoptosis, proliferation, angiogenesis, and metastasis [43,44]. Catenin delta-1 (CTNND1) functions as an oncogene and is known to be the driver of metastatic cancer progression in hepatocellular carcinoma, breast cancer, and colorectal cancer [45,46,47]. A very important gene that we identified is the six-transmembrane epithelial antigen of Prostate-2 (STEAP2), known to be over-expressed in aggressive PCa, which corroborates our study as it was identified in a high-grade tumor [48]. Furthermore, we also observed that the docking protein 5 (*DOK5*), a member of a subgroup of the DOK family, is known to be expressed using c-Ret in several neuronal tissues [49]. Recent studies have implicated its role in the invasion, progression, and metastasis of gastric cancer [50]. Amongst the lncRNAs, we identified that SOX2-OT is mapped to the chromosome locus 3q26.3 and is highly expressed in embryonic stem cells [51]. Deregulation of SOX2-OT is observed in various tumors, including lung cancer [52], gastric cancer [53], esophageal cancer [54], breast cancer [55], hepatocellular carcinoma [56], ovarian cancer [57], pancreatic [58], laryngeal squamous cell carcinoma, osteosarcoma, nasopharyngeal carcinoma, and glioblastoma [59,60]. The lncRNA FTX (five prime to XIST) possesses an X-inactive specific transcript and is involved in X-chromosome inactivation [61]. FTX has been reported to act as a tumor promoter in various types of cancer, including osteosarcoma [62], colorectal cancer [63], gliomas [64], lung adenocarcinoma [65], and gastric cancer [66], where it was found to be closely associated with a poor prognosis.

### 3.2. Protein Interactions Yielded Innate Pathways Responsible for PCa

The significant DEGs were used to build a gene interaction network using GeneMANIA and were later visualized using String and Cytoscape. The network was checked for the top-ranking genes using the expression correlation with a cut-off of 0.95 [67]. From the input genes shown with cross-hatched circles of uniform size, GeneMANIA added relevant genes which are shown with solid circles (hub genes are central), and their size is proportional to the number of interactions they have (Figure 3). In addition, we have identified several interacting partners that are co-expressed such as DZIP1, COL6A2, TAGLN, ZBTB33, LAMP1, LAMP2, IKBKB, and DNAJB11 which act as oncogenes in different cancers such as gastric cancer, renal cell carcinoma, and colorectal cancer, to name a few [68,69,70,71,72,73].

Finally, the network analyzer, Cytoscape-CytoHubba plugin was used to define the network measures, where yellow/orange (lower contrast) means the rank is lower and red/maroon (bigger contrast) indicates the rank is greater. We further evaluated the top network genes for clustering coefficient with the top 20 hub genes in our different malignant samples. Some of the prominent genes we identified through the networks are DOK5, APP, *CTNND1*, *STX6*, *STX10*, *STX16*, *BACE1*, and *BACE2* (Figure 4A–D); which, agree with the regulation of various cancers based on their co-expression patterns in GeneMANIA.

### 3.3. Validation of RNA-seq Result Using TCGA Dataset by Cbioportal

We sought to ask whether any DEGs were relatively expressed in publicly available datasets from various studies. To check this, we used TCGA datasets and checked for alteration of frequencies and expression in the cohorts. We used the PanCancer Atlas [74] dataset for Prostate Adenocarcinoma, where 489 samples were screened for their functional roles and molecular aberrations (Figure 5 and Figure 6. Mutations and CNV analysis for some of the DEGs were done and the results are summarized below:

Furthermore, an attempt was made to analyze genomic alterations such as gene amplifications, deep deletions (that are equivalent to homozygous deletions), shallow deletions (heterozygous loss), truncating mutations, in-frame mutations, or missense mutations. Among the DEGs, amplification was the most prominent one in the case of DOK5 and STEAP2; whereas, deep deletions and mutations were observed in COL6A1, STX6, and CTNND1. We argue that similar analysis could check the performance of all the DEGs which will highlight the alteration frequencies across the cohort (Figure 5).

### 3.4. Gene Ontology Yielded Distinct Pathways Regulating Biogenesis

All the DEGs obtained from our Cuffdiff pipeline were subjected to GO analysis by Pantherdb, which shows the role of DEGs in biological adhesions, biological regulations, biogenesis, cellular, metabolic and developmental processes, localization, locomotion, and multicellular organismal processes (Figure 7 and Figure 8).

For the molecular function category, the terms are binding factors, catalytic, adapter, transducer, and transcription regulator activities which are associated with differential expressed genes. This indicates the role of DEGs in different important processes like transcription, cell migration, differentiation, etc.

### 3.5. Phenolyzer Highlights Important DEGs

A cross-sectional comparison of phenotypes and DEGs using phenolyzer (http://phenolyzer.usc.edu last accessed on 7 April 2023; Figure 9) would provide us with indicators for the extent of expression across diseases [75]. Therefore, we asked how many DEGs among all, including the samples, cbioportal, and comparative analyses, are distinctly associated with disease/phenotype terms. Our Phenolyzer results with disease names, viz., Prostate Cancer, PCa neoplasia, which are clinical phenotype terms, have largely related to each other (seen in pink edges in Figure 8). On the other hand, to reach a consensus with the identification of DEGs, we also employed the DESeq/EdgeR normalization method which resulted in the identification of approximately 1230 genes as DEG: 490 were upregulated and 1215 were downregulated. We identified distinct genes including KLK4, FN1, PBOV1, TPM2, and FLNA which are known to be involved in PCa pathways along with IGF1, TPD52, and SRSF1 that are involved in different cancer pathways [76,77,78]. While KLK4 is a very important gene that is known to be involved in the progression of prostate cancer by promoting proliferation, migration, and epithelial to mesenchymal transition, we have checked its expression in our sample by qRT-PCR and we did see a significant change in benign vs. malignant samples. We also found a few novel lncRNAs such as LINC00940 and FLJ16779 that have not been reported earlier besides SNHG19, NPBWR1, and lncRNAs that are known to be involved in cancer and other diseases as well. A further attempt was made to understand the regulatory mechanisms underpinning PCa signature pathways by mapping lncRNAs with protein-encoding genes (Appendix A).

## 4. Discussion

In the current study, we have identified many DEGs which are known to be involved in different cancer pathways including PCa. While some of them are specifically known to be associated with PCa, we also discovered a few novel lncRNAs which need further investigation. Expression of COL6A1 is significantly elevated in different tumors such as lung, prostate, cervical, and pancreatic cancer compared to normal tissues. Interestingly, our previous WES studies in PCa have identified COL6A1 as one of the causal genes [9]; whereas we also identified this through our RNA-seq analysis. COL6A1 was shown to be physically interacting with DNAJB11, APP along with some other genes. DNAJB11 is involved in aberrant signaling pathways associated with different cancers. Similarly, APP is known to be associated with androgen-responsive genes and regulates the proliferation and migration of PCa cells [79]. Therefore, we argue that COL6A1 might act as a prognostic marker for PCa in the Indian population. Transcriptional factor Kaiso/ZBTB33 was identified as a CTNND1-specific binding partner and this complex is a modulator of the canonical Wnt/β-catenin signaling pathway [80]. There is a large amount of research focusing on the role of CTNND1 in cancer development and progression; however, in PCa, it is still not well-elucidated [81]. In our study, we have identified CTNND1 and ZBTB33 through interaction studies, but what is more interesting is that we have earlier identified CTNND1 as a co-localization partner with one of the lncRNAs (NONHSAT239888) which is known to be highly expressed in PCa [82]. Furthermore, CTNND1 is one of the main interacting partners of ACE2/TMPRSS2, the main receptors which are responsible for SARS-CoV-2 entry into the cell. We had hypothesized that CTNND1 interacts with EGFR, and this interaction could uphold the SARS-CoV-2 infection independent of its endocytosis and associated with cell viability [83]. The interacting partners of another important gene STEAP2 are KLK3, KLK2, and AR, all of which are hallmarks of PCa. KLK3 is a protein-coding gene, and its protein product, Prostate-specific antigen (PSA), is a well-established biomarker of PCa [84,85]. Similarly, human kallikrein 2 (KLK2), interacts with AR and drives PCa progression [86]. Earlier studies have shown that DOK5 are expressed in T-cells and their expression is regulated upon T-cell activation. DOK5 is shown to be involved in the invasion and metastasis of cancer specifically in gastric cancer, but it has not been well-studied in PCa [49]. Since we got a strong interaction of DOK5 in our clustering coefficient studies using the cytoHubba plugin, we argue that it would be worth analyzing this gene further. Interestingly, through our cytoscape-cytoHubba analysis, we identified many soluble *N*-ethylmaleimide-sensitive factor (NSF) attachment protein receptor (SNARE) proteins that are key mediators of membrane fusion [87]. All the SNARE proteins share a common sequence of 60–70 residue called the SNARE motifs, which helps in mediating the interaction between vesicle SNAREs (v-SNAREs) and targetting membrane SNAREs (t-SNAREs) [88]. One of the t-SNARE proteins, syntaxin 6 (STX6), is particularly important in vesicle fusion. STX6 is upregulated in a variety of cancers including breast, colon, liver, pancreatic, prostate, bladder, skin, testicular, tongue, cervical, lung, and gastric cancers, and it has been identified as a common transcriptional target of the p53 family members (p53, p63, and p73) [89]. Along with STX6, we have identified STX10, VTI1A, and STX16 which can be further studied. Overall, we also found other important DEGs, viz., BACE1 and BACE2, which belong to a class of proteases called β-secretases that are extensively studied in Alzheimer’s disease [90]. Not many studies have been done with respect to their role in cancer, but there are some recent studies that have shown their involvement in pancreatic and skin cancers [91].

### 4.1. Comparative Analysis of RNA-seq Data with Other Publicly Available Datasets

To screen the potential DEGs across the datasets, we compared and analyzed the DEGs from our current study to those of the DEGs from cbioportal and methylation/array-specific datasets that are publicly available from NCBI gene expression omnibus (GEO; https://www.ncbi.nlm.nih.gov/geo/ last accessed on 13 April 2023). As no concrete list of RNA-seq datasets was associated in the sequence read archive in lieu of PCa phenotype, we compared our DEGs to GEO datasets, viz., *GSE6919* and *GSE45016,* in addition to the previously benchmarked RNA-seq dataset of PCa in the Chinese population [92]. From this, we obtained only 135 hits in SRA for the RNA-seq of PCa, and most datasets are either from PCa cell lines or RIP-seq; hence, we deemed them not useful keeping in view of diffident phenotypes, experiments, and perhaps correlation studies that they may be heralded with. Nevertheless, we identified a few common genes which could be the key candidate genes in our study, viz., STEAP2, DOK5, Il6ST, LMOD1, CTNND1, etc. Likewise, when our data were compared with GEO datasets, we observed IL6ST, BACE2, SOX2-OT, STEAP2, APP, SNX9, STX16, and CTNND1 among the other genes that are expressed. This, we believe, strengthens our finding that the DEGs which we have mentioned in the current study could be a valid signature for PCa diagnosis. On the other hand, when we compared Prostate Adenocarcinoma (TGCA, PanCancer Atlas) dataset to that of our list of DEGs, we found IL6ST and ZBTB20. These key DEGs are common between the Indian population and the Western population, which can be validated further as it is beyond the scope of this current analysis.

### 4.2. A Major Chunk of lncRNAs Are Novel and Regulated in Distinct Pathways

LncRNAs do not code for proteins, but they are involved in almost all biological processes such as gene expression, epigenetic regulation, cell cycle regulation, etc., in different cancers, PCa being one of them. Previous studies have highlighted the oncogenic role of lncRNAs in metastasis, proliferation, and development of PCa, but still there are several lncRNAs whose functions are still unknown. With the advancement in NGS, bioinformatics analysis has enabled the identification of many lncRNAs which show dysregulated expression in PCa [93]. Their diverse role has made them a target for all stages of PCa development which includes screening, diagnosis, prognosis, and treatment, further establishing their role as biomarkers in PCa. For example, MALAT-1 (Metastasis associated lung adenocarcinoma transcript 1), a lncRNA, which is used to predict metastasis and survival in non-small cell lung cancer [94], has also been correlated with PCa development and progression [95]. It has also been reported that MALAT-1 expression closely correlates with PSA levels, Gleason scores, and tumor sizes [93]. Similarly, PCAT-18 (prostate-cancer-associated non-coding RNA transcript 18 and SChLAP1 (second chromosome locus associated with prostate-1)) has also been used as a diagnostic and prognostic biomarker in PCa [96,97]. Some of the lncRNAs we identified from our current study include LINC01440, SOX2OT, ENSG00000232855, ENST00000647843.1, and FTX. FTX has been reported to be involved in the tumorigenesis of multiple cancer types. Long intergenic non-protein coding RNA 1440 (LINC01440) is a novel lncRNA that needs to be explored for its role in cancers. A recent study has implicated its role in a spinal disorder known as Ossification of ligamentum flavum (OLF), where it was found to be upregulated in the diseased patients compared to the healthy population [98]. Different studies have shown that SOX2OT acts as an oncogene and is elevated in different tumor types. However, its role and significance are still not explored in PCa, which makes it a very promising candidate for further analysis. Another lncRNA, ENSG00000287903 (NONHSAT106693), was earlier screened in our Vitamin K deficiency cohort from our recent study [99]; wherein, qRT-PCR showed a significant upregulation in malignant samples compared to control samples (Appendix A), which provides us a raison d’être to further validate the remaining lncRNAs as well. Given the role of lncRNAs in PCa, it would be interesting to see if any of these lncRNAs can serve as biomarkers; albeit several downstream experiments and well-designed clinical trials are to be employed. Taken together, there are a few limitations to our study. (i) The qRT-PCR validation needs more tumor and adjacent normal tissue samples as our sample size is very small. (ii) Some additional experiments, such as immunohistochemistry and Western blot, when performed, could confirm the protein levels in PCa. (iii) Given the scarcity of PCa data in India, the survival analysis across the entire transcriptome could not be drawn as it is largely towards the identification of biomarkers indicating the prognostic power, but given the sample size conundrum we have, we are limited in checking this power. Nevertheless, in our previous study, we performed WES on PCa samples, which we have cited, but again, the sample size is limited. (iv) Due to a lack of fresh–frozen prostate/radical prostatectomy tissues, only FFPE blocks were used, and isolating RNA from them is an arduous and challenging task [39]; due to this, we could not perform qRT-PCR for all the selected DEGs.

## 5. Conclusions

Prostate cancer cases are increasing in India even as NGS studies are just beginning to be explored. In our current RNA-seq study and subsequent bioinformatics analyses, we sought to characterize DEGs including lncRNAs that are specific to PCa of an Indian sub-population. While we identified some of the important genes, viz., *DOK5*, *COL6A1*, *CTNND1*, *STEAP2*, and *APP,* their role in PCa is still not clear. We envisage that characterizing their functional aspects would help us understand PCa progression. Since the search for non-invasive and more sensitive biomarkers is on the anvil across all solid tumors, we firmly hope that these lncRNAs amongst the DEGs would serve as a precedent in the development of NGS panels for PCa detection in the Indian phenotype.

## Figures and Tables

**Figure 1 diseases-11-00072-f001:**
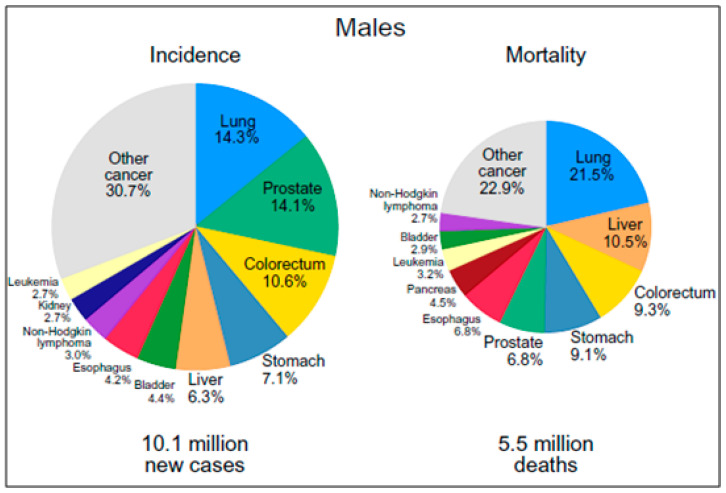
The incidence and mortality rates for the top 10 most common cancers of 2020 (adapted from Globocan 2020).

**Figure 2 diseases-11-00072-f002:**
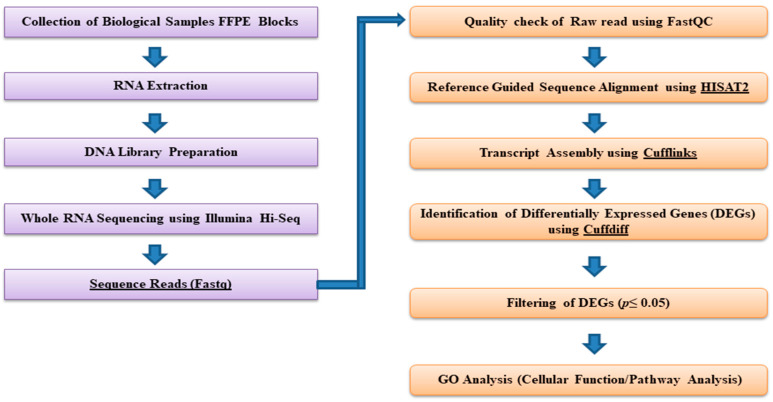
Pictorial representation of the RNA-seq workflow depicting different steps of analysis. The input consists of FASTQ files of the sample (control and malignant), human reference genome sequence file (hg38), and gene annotations from gtf files.

**Figure 3 diseases-11-00072-f003:**
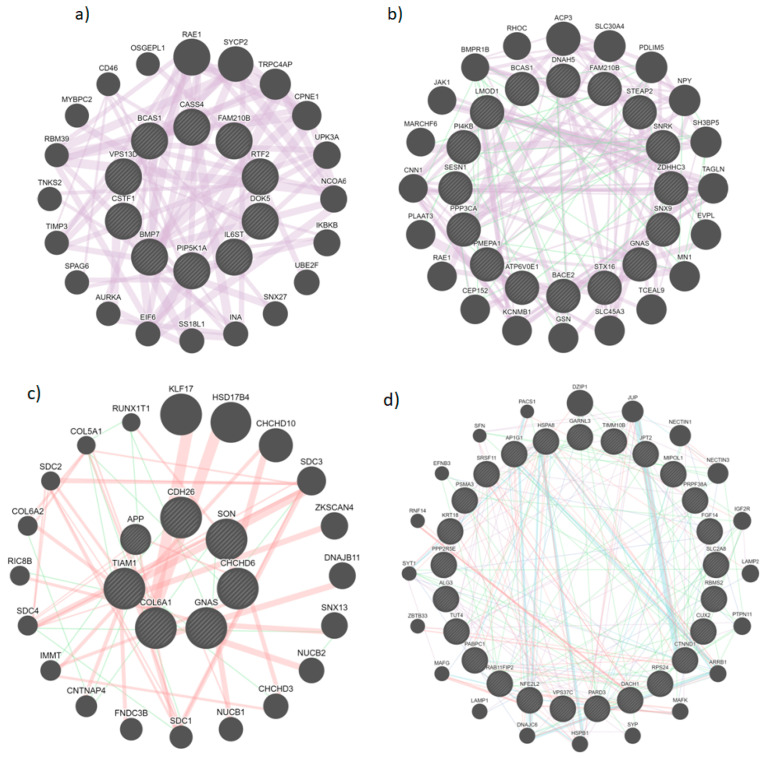
Protein—protein interaction of different samples using GeneMANIA with purple edges representing co-expression; green—genetic interaction; red—physical interaction; and blue representing co-localization. ((**a**–**d**) represents different sample pairs (with #1374 as a control) from which DEGs were inferred, viz., 69/19, 1631/H19, 4226/H19, 5110/H20, respectively, and the circles with edges are the input genes whereas solid circles are the interacting partners).

**Figure 4 diseases-11-00072-f004:**
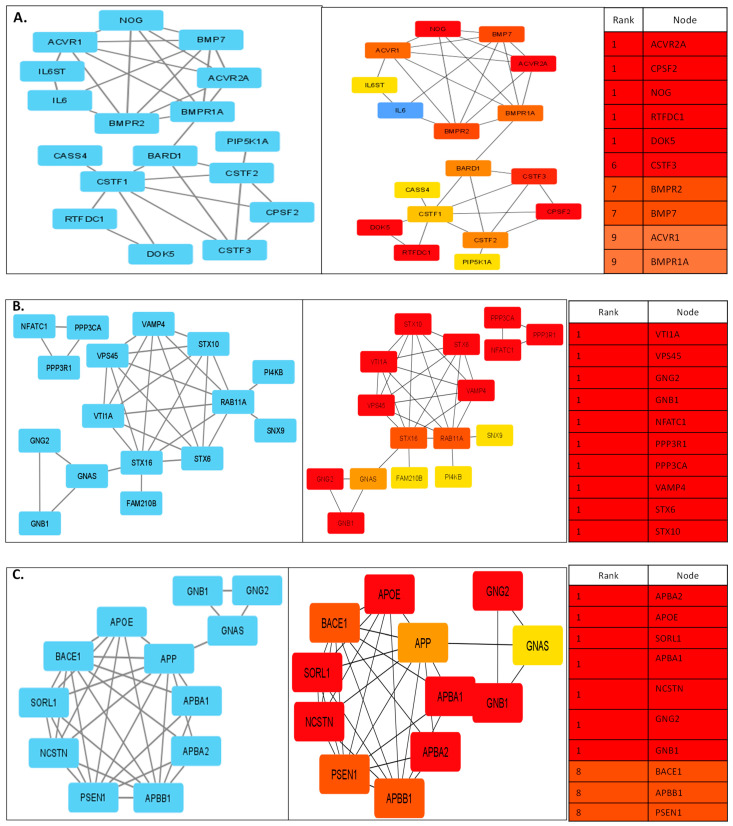
A compendium of the top 20 genes from the PPI network that are regulated and expressed in malignant samples was considered using the Cytoscape-CytoHubba plugin. The network was ranked based on clustering coefficients which yielded hierarchically high confidence interactions. These genes with the highest clustering coefficients are indicated in red, while high to moderate clustering coefficients are in orange and those with low clustering coefficients are shown in yellow. Some of the important genes we identified through all 4 networks ((**A**–**D**) represents 4 different sample pairs from which DEGs were analyzed) are *DOK5*, *APP*, *CTNND1*, *STX6*, *STX10*, *STX16*, *BACE1*, and *BACE2*, which are amongst the top-ranking genes in the network. It is imperative that some genes are ranked based on the clusters they make.

**Figure 5 diseases-11-00072-f005:**
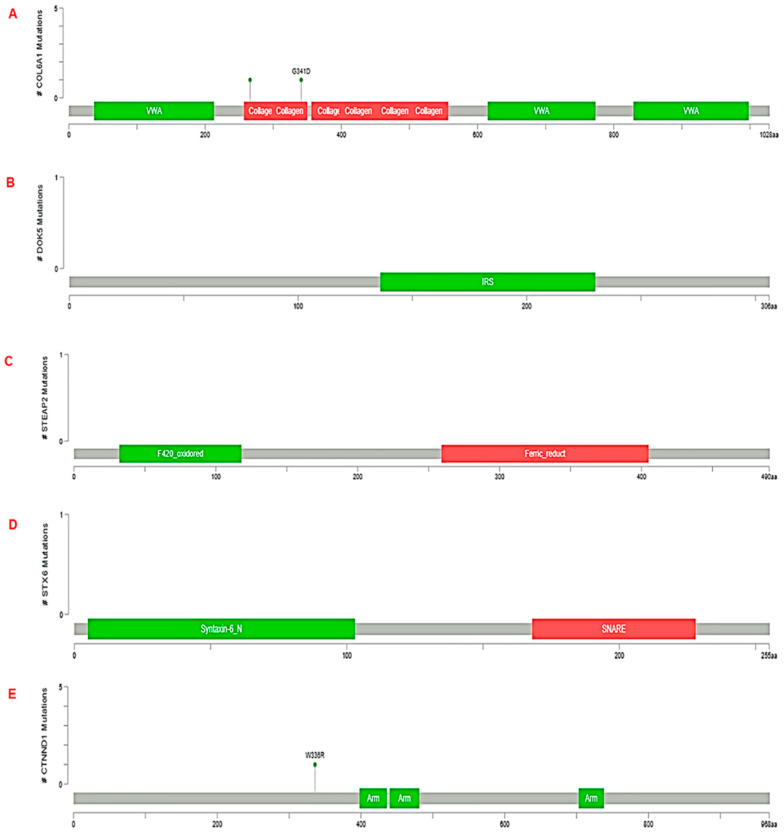
Graphical summary of corresponding genes with mutation types. We employed the cBioportal and analyzed different mutation types in genes; (**A**) *COL6A1*, (**B**) *DOK5*, (**C**) *STEAP2*, (**D**) *STX6*, and (**E**) *CTNND1* related to PCa by comparing with integrated genomic data of different alterations available in it. Different color codes depict different mutation types. Here, green shows missense mutations of unknown significance, grey for truncating mutations, and red for in-frame putative mutations.

**Figure 6 diseases-11-00072-f006:**
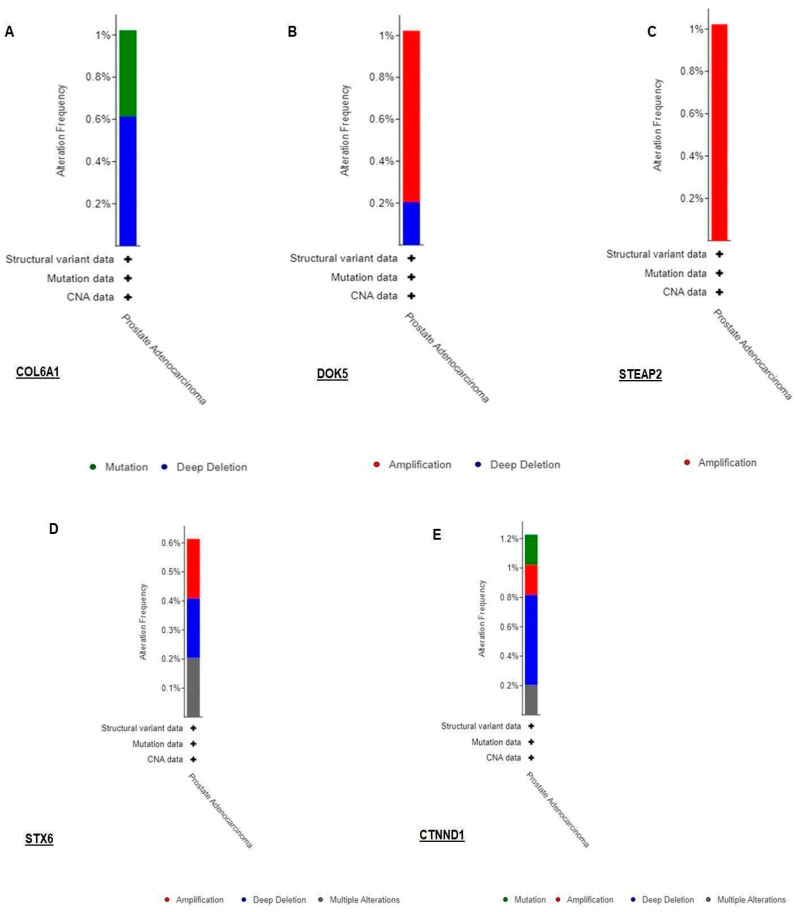
Graphical representation of gene expression and alteration frequency. (**A**) COL6A1, (**B**) DOK5, (**C**) STEAP2, (**D**) STX6, and (**E**) CTNND1 were queried using an available gene expression database of Prostate adenocarcinoma cancer type in the cBioportal tool where the different colors green, red, blue, and grey code mutations, amplifications, deep deletion, and other multiple alterations, respectively. The queried gene is altered either in 1% or less than 1% of queried patients/samples (total number of samples—489).

**Figure 7 diseases-11-00072-f007:**
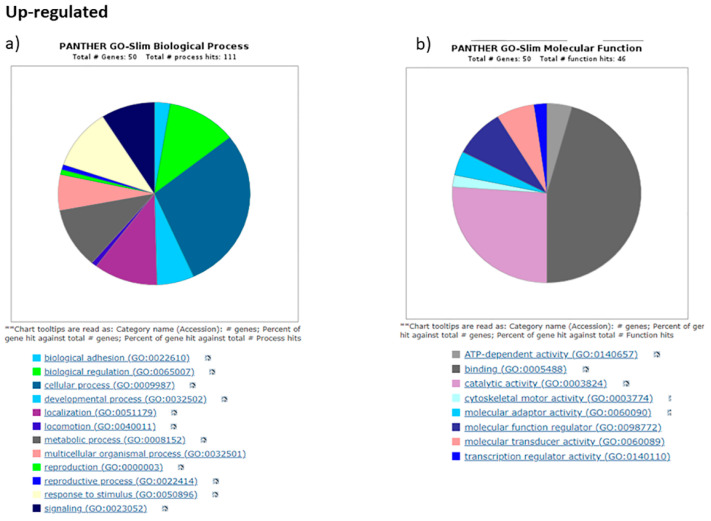
Gene Ontology analysis using Pantherdb. We used GO database for upregulated DEGS from our study, with the genes clustered based on biological functions (BP) and Molecular functions (MP). Important pathways associated with them are listed below corresponding to different colors.

**Figure 8 diseases-11-00072-f008:**
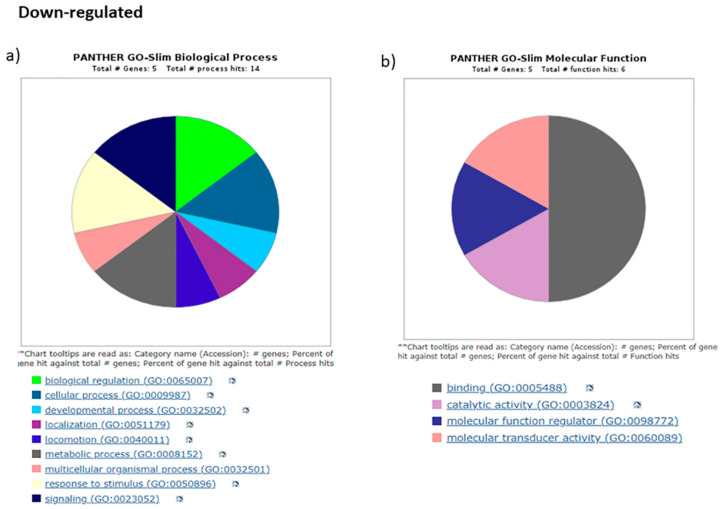
Gene Ontology analysis using Pantherdb. We used GO database for downregulated DEGS from our study with the genes clustered based on biological functions (BP) and Molecular functions (MP). Important pathways associated with them are listed below corresponding to different colors.

**Figure 9 diseases-11-00072-f009:**
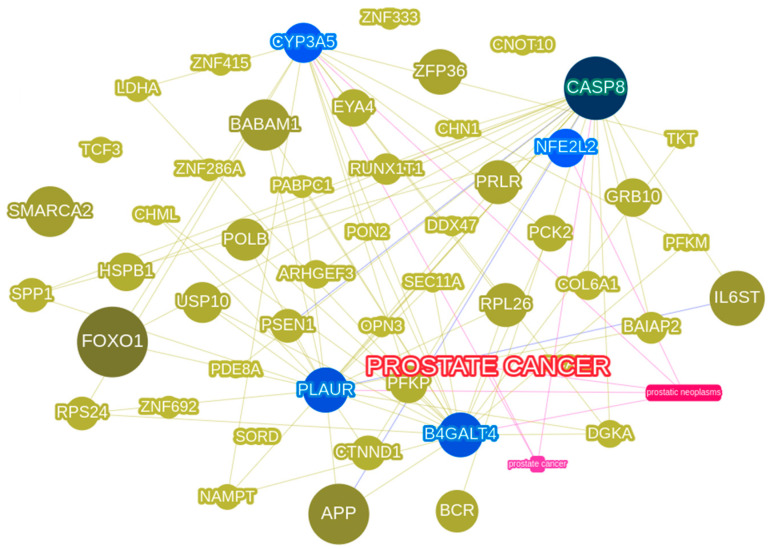
Phenolyzer hub of DEGs associated with distinct clinical terms, viz., PCa and PCa neoplasia.

**Table 1 diseases-11-00072-t001:** Samples used for the WTSS along with their Gleason scores constituted two high-grade tumor samples with two intermediate samples, while 2 others with less than 6 from benign cases.

Sample ID	Condition	Gleason Score (Primary + Secondary)
69/19	Adenocarcinoma	3 + 4
1631/H19	Adenocarcinoma	4 + 4
4226/H19	Adenocarcinoma	3 + 4
5110/H20	Adenocarcinoma	4 + 4
1374/19	Benign Nodular Prostatic Hyperplasia	<6
1266/19	Benign Nodular Prostatic Hyperplasia	<6

**Table 2 diseases-11-00072-t002:** The criteria used for inclusion and exclusion of the test subjects.

	Inclusion Criteria	Exclusion Criteria
Malignant (PCa)	Age > 55 years PSA > 4 ng/mL Gleason score > 6 Non-diabetic or any other co-morbidity	Smoking Familial history of BPH
Normal (BPH)	Age > 55 years PSA < 4 ng/mL Gleason score < 6	Those with Associated diseases/phenotypes or any urological diseases.

**Table 3 diseases-11-00072-t003:** List of DEGs obtained with the designated cut-off after Cufflinks-Cuffdiff analyses. Some of the identified DEGs were PCa-specific such as APP, STEAP2, PABPC1, RPS24, etc.

Gene Id	Locus	log2 Fold	*p*-Value	Gene Name
CUFF.1000	chr1:12314982–12315401	3.1420	0.0254	VPS13D
CUFF.100001	chr20:53952153–53952369	2.9034	0.0366	BCAS1
CUFF.10004	chr1:151206377–151206638	4.6543	0.02435	PIP5K1A
CUFF.100042	chr20:54490801–54490981	8.8744	0.0245	DOK5
CUFF.100060	chr20:55553908–55554084	7.1775	0.0213	LINC01440
CUFF.134059	chr5:55937823–55938153	4.6464	0.0334	IL6ST
CUFF.100106	chr20:57168914–57169100	2.8468	0.02305	BMP7
CUFF.100077	chr20:56403670–56403826	2.7896	0.011	CSTF1
CUFF.100080	chr20:56412720–56412894	3.2345	0.02435	CASS4
CUFF.100086	chr20:56487111–56487297	3.1269	0.02305	RTF2
CUFF.102595	chr21:41275383–41275947	3.4149	0.0307	BACE2
CUFF.109200	chr3:43335859–43336390	3.4173	0.0307	SNRK
CUFF.109337	chr3:44926095–44926812	4.0190	0.02435	ZDHHC3
CUFF.119448	chr3:181441034–181441358	3.3563	0.0131	SOX2OT
CUFF.126886	chr4:101280781–101281251	3.0588	0.0234	PPP3CA
CUFF.131979	chr5:13991457–13991753	3.1847	0.04515	DNAH5
CUFF.13343	chr1:201898030–201898394	4.054158	0.02435	LMOD1
CUFF.141632	chr5:173034434–173034832	4.094679	0.02855	ATP6VOE1
CUFF.151326	chr6:157942571–157943399	4.254992	0.004	SNX9
CUFF.157342	chr7:90232992–90235254	3.248	0.03545	STEAP2
CUFF.100128	chr20:57648598–57649155	4.119627	0.00535	PMEPA1
CUFF.10017	chr1:151326615–151327193	3.3082	0.0272	PI4KB
CUFF.100074	chr20:56367973–56368588	4.4636	0.0028	FAM21OB
CUFF.100241	chr20:58895335–58895591	2.7206	0.0227	GNAS
CUFF.100175	chr20:58033916–58034202	5.059	0.04605	STX16
CUFF.100056	chr20:55074569–55074768	2.743	0.00795	RPL12P4
CUFF.100291	chr20:59967544–59968164	3.415	0.04675	CADHERIN 26
CUFF.100698	chr21:14084939–14085128	3.123919	0.0416	ENSG00000224905
CUFF.10107	chr1:152032497–152032839	4.3215	0.0145	ENSG00000229021
CUFF.101153	chr21:18857730–18858445	2.8043	0.04035	PPIAP22
CUFF.101277	chr21:25880415–25881777	5.6321	0.03945	APP
CUFF.101659	chr21:28672882–28673291	3.3034	0.00035	ENSG00000232855
CUFF.101853	chr21:31210389–31210838	3.2157	0.0433	TIAM1
CUFF.102028	chr21:33551193–33553146	5.4886	0.02575	SON
CUFF.102981	chr21:46003384–46005044	7.4652	0.0004	COL6A1
CUFF.115488	chr3:126861928–126865263	3. 9928	0.04385	CHCM1/CHCHD6
CUFF.101498	chr8:100713839–100714204	3.23209	0.03765	PABPC1
CUFF.110085	chrX:74232840–74233182	3.23919	0.03765	FTX
CUFF.103580	chr9:14087901–14088194	3.42888	0.04005	NFIB
CUFF.2892	chr1:52416916–52417541	3.13921	0.0423	TUT4
CUFF.41953	chr17:48060789–48061102	4.18946	0.04285	NFE2L2
CUFF.36847	chr16:1699878–1700192	3.07352	0.044	HN1L/JPT2
CUFF.22947	chr12:52949238–52950383	3.04539	0.04445	KRT18
CUFF.15787	chr11:6483526–6483970	3.31699	0.04535	TIMM10b
CUFF.3812	chr1:70224556–70224999	3.31713	0.04535	SRSF11
CUFF.20358	chr11:123057501–123061280	4.93401	0.0459	HSPA8
CUFF.30773	chr14:58259607–58260234	2.96918	0.046	PSMA3
CUFF.39071	chr16:71729367–71729916	2.97406	0.046	AP1G1
CUFF.14995	chr10:118005601–118006215	3.17633	0.049	RAB11FIP2
CUFF.17593	chr11:61130859–61131197	2.90619	0.0491	VPS37C
CUFF.11546	chr10:34656519–34656811	3.1245	0.00075	PARD3
CUFF.31039	chr14:63395971–63396275	2.8375	0.00075	PPP2R5E
CUFF.110851	chrX:111699308–111699657	3.8764	0.00085	ALG3
CUFF.25323	chr12:111257228–111257830	3.8654	0.00095	CUX2
CUFF.107185	chr9:127397206–127397465	3.2564	0.00105	SLC2A8
CUFF.107177	chr9:127325581–127325969	4.3465	0.0007	GARNL3
CUFF.17462	chr11:57782675–57782947	4.93401	0.0003	CTNND1
CUFF.13238	chr10:78033882–78040677	6.18617	0.0252	RPS24
CUFF.100491	chr8:71098236–71098451	7.9462	0.00075	ENST00000647843.1
CUFF.55161	chr15:44826300–44826876	5.03278	0.0428	ENST00000558419.1
CUFF.10014	chr1:151287575–151287979	3.9867	0.01955	ZNF687
CUFF.100478	chr20:63890280–63890538	4.8968	0.0235	TPD52L2
CUFF.100559	chr21:6986631–6987286	2.9645	0.0209	ENST00000623165.3
CUFF.103593	chr22:22900959–22901440	4.6579	0.01695	IGLC2
CUFF.100039	chr8:60846260–60846506	3.8965	0.03515	CHD7
Down-regulated				
CUFF.148166	chr6:109038801–109039233	−2.85649	0.0181	SESN1
CUFF.27895	chr13:71866547–71867205	−4.47892	0.03305	DACH1
CUFF.28742	chr13:102161569–102161764	−3.74264	0.00095	FGF14
CUFF.30017	chr14:37573929–37574220	−2.98346	0.001	MIPOL1
CUFF.23189	chr12:56596124–56596418	−5.36542	0.0008	RBMS2

## Data Availability

The data that support the findings of this study are available at the National Center for Biotechnology Information (NCBI) Sequence Read Archive (SRA) database; accession number-PRJNA616165.

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
