# Peer review of "Towards Understanding the Key Signature Pathways Associated from Differentially Expressed Gene Analysis in an Indian Prostate Cancer Cohort"

_diseases, 2023, doi:10.3390/diseases11020072_

Round 1

Reviewer 1 Report

In this work, Shukla et al. proposed a work to identify DEGs and long non-coding RNAs associated with signature pathways from an Indian Prostate cancer cohort using from RNA-seq data. In the results several important genes known to be involved in different cancer pathways are reported as well as a few novel lncRNAs. I believe the bioinformatics analyses in this work are solid, and a few more results can be added to improve the overall quality of the work as follows:

1.      It is interesting to see the pathway analysis from the RNA-Seq data, however, the manuscript only mentioned clustering from Pantherdb GO databases. Can the author add enrichment analysis for known pathways, for example, with KEGG pathway database?

2.      For the analysis results of mutation types with respect to corresponding genes, can structural prediction be added to validate the effect of mutations, especially for the missense mutations of unknown significance mentioned in section 3.3?

3.      It seems some figures have an issue with resolutions. For example, the texts in Figures 6 and 7 are hard to read.

Author Response

Reviewer 1

In this work, Shukla et al. proposed a work to identify DEGs and long non-coding RNAs associated with signature pathways from an Indian Prostate cancer cohort using from RNA-seq data. In the results several important genes known to be involved in different cancer pathways are reported as well as a few novel lncRNAs. I believe the bioinformatics analyses in this work are solid, and a few more results can be added to improve the overall quality of the work as follows:

  1.     It is interesting to see the pathway analysis from the RNA-Seq data, however, the manuscript only mentioned clustering from Pantherdb GO databases. Can the author add enrichment analysis for known pathways, for example, with KEGG pathway database?

Thank you for your suggestion. We have added a supplementary figure for KEGG analysis. 

  1.     For the analysis results of mutation types with respect to corresponding genes, can structural prediction be added to validate the effect of mutations, especially for the missense mutations of unknown significance mentioned in section 3.3?

Thank you for your excellent points. We checked for inherent mutations at the interfacial sites with PDB and found no mutations that were mapped from cbioportal. We don’t have a structure in PDB for COL6A1.  For DOK5, there are no pathogenic mutations found at the interfacial site.  

  1.     It seems some figures have an issue with resolutions. For example, the texts in Figures 6 and 7 are hard to read.

Thank you. We have tried to increase the resolution.  The resolution in doc is lower, but we have them in figures and supp figures

Reviewer 2 Report

The work of Shukla et al is interesting, but I do not recommend publication in the current form. The raw data for identifying differentially expressed genes (DEGs) is not available and the statistical analysis of the findings is not presented properly. In general, the authors present their interpretation of the findings instead of the actual findings. As a result, a reader cannot make an opinion on how robust the interpretations are and therefore this article is not useful as is. Specifically,

1.       How many DEG calculations were performed? The authors should provide the data in a supplemental table, together with the log2 fold and p-values.

2.       Was there any correction for multiple hypotheses? If not, why?

3.       Table 2 shows part of the identified DEGs. What are the rest? Why only these are shown? How they were ordered?

4.       For Figure 2, it is unclear what A, B, C, and D panels correspond to. Why these genes/interactions are shown and what is the conclusion of the analysis? Is there an interaction enrichment in DEG with high p-values compared to random p-values?

5.       Similarly, it is hard to follow what the hypothesis, test, and result is in the analysis shown in Figure 3. How they were defined and which are the “top network genes” and the “top 20 hub genes”?

6.       What is the outcome of the analysis using TCGA dataset by cbioportal? How this helps the reader? I could say the same for the rest figures (Fig 4-8) of the manuscript.

7.       The authors should also mention the high prevalence of PCa in men of African ancestry.

8.    The inclusion and exclusion criteria for PCa patients need brief justifications. Also, clear presentation (i.e. normal BPH was included or excluded?).

Author Response

Reviewer 2

The work of Shukla et al is interesting, but I do not recommend publication in the current form. The raw data for identifying differentially expressed genes (DEGs) is not available and the statistical analysis of the findings is not presented properly. In general, the authors present their interpretation of the findings instead of the actual findings. As a result, a reader cannot make an opinion on how robust the interpretations are and therefore this article is not useful as is. Specifically,

  1.       How many DEG calculations were performed? The authors should provide the data in a supplemental table, together with the log2 fold and p-values.

Thank you. The DEGs were selected based on the p-value less than or equal to 0.05. We have given all the DEGs in the supplementary file. 

  1. Was there any correction for multiple hypotheses? If not, why?

Thank you. We performed the DEG analysis using Cufflinks and checked for p-value heuristics and filtered the DEG profiles based on 2>=log2FC<=2.  The q value was considered in these cases to filter and check the upregulated and downregulated genes.

  1.       Table 2 shows part of the identified DEGs. What are the rest? Why only these are shown? How they were ordered?

Thank you. Now, we have included all the genes in table 2 (though earlier we had given that information in supplementary table)

  1.       For Figure 2, it is unclear what A, B, C, and D panels correspond to. Why these genes/interactions are shown and what is the conclusion of the analysis? Is there an interaction enrichment in DEG with high p-values compared to random p-values?

Thank you. The panels A, B, C and D correspond to four different malignant samples (A-69/19, B-1631/H19, C-4226/H19, D-5110/H20). GeneMANIA is a versatile, user-friendly web tool for developing gene function hypotheses, reviewing gene lists, and ranking genes for functional experiments. Through this, we identified several interacting partners which are also involved in different cancer pathways, which gave us a strong reason for further analysis of these genes. 

  1.     Similarly, it is hard to follow what the hypothesis, test, and result is in the analysis shown in Figure 3. How they were defined and which are the “top network genes” and the “top 20 hub genes”?

Thank you. When we queried all the DEGs across the samples, we chose the top 10 DEGs based on clustering coefficient using Cytohubba app in Cystocape.  

  1.       What is the outcome of the analysis using TCGA dataset by cbioportal? How this helps the reader? I could say the same for the rest figures (Fig 4-8) of the manuscript.

The TCGA is a gold standard dataset for retrieving and identifying causal genes in cancers. We were intrigued to check whether or not there are any DEGs in lieu of PCa pathogenesis in Cbioportal. Hence we have considered Cbioportal.  

  1.       The authors should also mention the high prevalence of PCa in men of African ancestry.

Thank you. We have added a brief para on this, 

“Prostate cancer (PCa) is the fifth most common cause of cancer death in males worldwide and the second most frequently diagnosed malignancy in many developed nations. Notwithstanding the increased burden of PCa incidence in affluent nations, males of African ancestry (MAA) bear a disproportionate amount of this burden. Men living in the Caribbean and Sub-Saharan Africa had the highest PCa mortality rates in the world. According to the International Agency for Research on Cancer (IARC), PCa is also anticipated to trend higher across Africa, where fatalities from the disease will rise from about 28,000 in 2010 to just over 57,000 by 2030.”

  1.   The inclusion and exclusion criteria for PCa patients need brief justifications. Also, clear presentation (i.e. normal BPH was included or excluded?).

Thank you. Now, we have added one table for patient inclusion and exclusion criteria. 

Reviewer 3 Report

In the manuscript entitled “Towards Understanding the Key Signature Pathways Associated from Differentially Expressed Gene Analysis in an Indian Prostate Cancer Cohort”, the authors identified DEGs as well as long non-coding RNAs (lncRNAs) associated with signature pathways
from an Indian PCa cohort with RNA-sequencing approach. The subject is really interesting and fascinating for publication, but there are some comments regarding the manuscript.
Authors may find useful to consider following comments and suggestions in preparation of the manuscript. Nevertheless, I believe the paper can be accepted for publication.

In the introduction section, there must be figure, graphic etc that reflects prostate cancer occurrence.

Overally, the manuscript has some punctuation errors and needs to be corrected.

Figure 1 should be simplified.

There are too many references.

Thoroughly check the consistency of references.

Author Response

Reviewer 3

In the manuscript entitled “Towards Understanding the Key Signature Pathways Associated from Differentially Expressed Gene Analysis in an Indian Prostate Cancer Cohort”, the authors identified DEGs as well as long non-coding RNAs (lncRNAs) associated with signature pathways from an Indian PCa cohort with RNA-sequencing approach. The subject is really interesting and fascinating for publication, but there are some comments regarding the manuscript. Authors may find useful to consider following comments and suggestions in preparation of the manuscript. Nevertheless, I believe the paper can be accepted for publication.

1)In the introduction section, there must be a figure, graphic etc that reflects prostate cancer occurrence.

Thank you, we have added one figure about prostate cancer occurrence (Globocan, 2020)

2)Overall, the manuscript has some punctuation errors and needs to be corrected.

Thank you. We have tried to modify that. 

3)Figure 1 should be simplified.

Thank you. We have added some more steps in the flow chart.

4) There are too many references.

Thank you for your point. We have added a few references as other reviewers suggested us to do so. Hope we may thankfully decline this comment. 

5)Thoroughly check the consistency of references.

Thank you for your suggestions. We have tried to include all your suggestions in our manuscript. 

Reviewer 4 Report

In this study, the authors used RNA sequencing to identify differentially expressed genes (DEGs) and long non-coding RNAs (lncRNAs) associated with signature pathways in an Indian prostate cancer cohort. They identified important genes specific to PCa and some novel lncRNAs that need further characterization. This study sets a precedent for further experimental validation and may pave the way toward the discovery of biomarkers and the development of novel therapies for PCa.

 The research is of particular interest to the scientific community and the work is supported by experimental data. However, this referee believes that the results sections should be implemented to guide the reader in understanding the results.

- First of all, the authors present the results in an organized way, but a major English language revision is necessary to better help the audience understand the scope and outcomes of the work. Please double-check plurals, verb conjugations and past tenses, sentence structures, pronouns, articles, etc.

- The abstract, apart from the minor inaccuracies listed below, is well-written and reflects the content of the article.

- For Table 1 it would be better to include the patient selection criteria to guide the reader to slowly familiarize with the subject.

- In the supplementary information of RT-qPCR it should be nice to indicate how many times you performed each experiment and the statistical significance of the analysis.

- The sentence “These tools have gained worldwide attention and have been used in a number of transcriptomics studies” could be reworded to make it less generic and more specific. For example, it could be indicated which were the most relevant results obtained through the use of these tools in other transcriptomics studies.

- The sentence “To check this, GeneMANIA was employed, which overcomes the limitations of earlier methods for predicting gene function on yeast and animal benchmarks using GeneMANIA (https://genemania.org/ last accessed on December 26, 2022)” is a bit confusing. It could be reworded for greater clarity.

- The sentence “The statistics for the aforementioned analyses were based on network analyses with every approach having a function F attached to it that gives each node v a numerical value (v)” is a little redundant and could use rephrasing.

- In the sentence “With ensuing cufflinks-cuffdiff pipeline, we obtained approximately 70 DEGs among which 65 were up-regulated and 5 were down-regulated with an inherent p-value heuristics <=0.05 and <=-2 Log2FC >=Log2FC” it might be useful to specify better the criteria used to identify DEGs. For example, it could be explained in more detail how the p-values and Log2FC were calculated and their biological meanings.

Bibliographic data in the reference list needs a thorough review.

Author Response

Reviewer 4

In this study, the authors used RNA sequencing to identify differentially expressed genes (DEGs) and long non-coding RNAs (lncRNAs) associated with signature pathways in an Indian prostate cancer cohort. They identified important genes specific to PCa and some novel lncRNAs that need further characterization. This study sets a precedent for further experimental validation and may pave the way toward the discovery of biomarkers and the development of novel therapies for PCa.

 The research is of particular interest to the scientific community and the work is supported by experimental data. However, this referee believes that the results sections should be implemented to guide the reader in understanding the results.

1) First of all, the authors present the results in an organized way, but a major English language revision is necessary to better help the audience understand the scope and outcomes of the work. Please double-check plurals, verb conjugations and past tenses, sentence structures, pronouns, articles, etc.

Thank you for your suggestion. We have tried to improve it. 

2) The abstract, apart from the minor inaccuracies listed below, is well-written and reflects the content of the article.

Thank you.

3) For Table 1 it would be better to include the patient selection criteria to guide the reader to slowly familiarize with the subject.

Thank you for your suggestion. We have included one table for patient inclusion and exclusion criteria.

Inclusion criteria

Exclusion criteria

Malignant (PCa)

Age>55 years

PSA>4ng/ml

Gleason score>=6

Non-diabetic or any other co-morbidity

Smoking

Familial history of BPH

Normal (BPH)

Age>55 years

PSA<4ng/ml

Gleason score<6

Those with Associated diseases/phenotypes or any urological diseases. 

4) In the supplementary information of RT-qPCR it should be nice to indicate how many times you performed each experiment and the statistical significance of the analysis.

Thank you for your suggestion. We have added some more information;

The real time qRT-PCR was performed in the LightCycler 480 II (Roche) using Applied Biosystems™ Power SYBR™ Green PCR Master Mix (ThermoFisher Scientific, Cat No-4367659) in a total volume of 10µl. Three biological replicates were used for all of the reactions, and LightCycler® 480 software (1.5.1) was used to determine the threshold cycle (Ct). The relative quantitative expression level was calculated by the 2−∆∆CT method (Livak and Schmittgen, 2001). Each experiment was repeated twice. 

5) The sentence “These tools have gained worldwide attention and have been used in a number of transcriptomics studies” could be reworded to make it less generic and more specific. For example, it could be indicated which were the most relevant results obtained through the use of these tools in other transcriptomics studies.

Thank you. We have changed this line, now it reads as

Cufflink-Cuffdiff has been used for transcriptome analysis in a wide number of studies for e.g. Tripathi et al have identified differentially expressed genes from breast cancer. Similarly, Kim et al have  done transcriptome analysis of sinensetin-treated liver cancer cells using cufflinks-cuffdiff pipeline. There are many more studies which corroborates the importance of these tools.

6) The sentence “To check this, GeneMANIA was employed, which overcomes the limitations of earlier methods for predicting gene function on yeast and animal benchmarks using GeneMANIA (https://genemania.org/ last accessed on December 26, 2022)” is a bit confusing. It could be reworded for greater clarity.

Thank you. Now, we have changed this line to-

To check this, GeneMANIA (https://genemania.org/ last accessed on December 26, 2022) was employed which is a versatile, user-friendly web tool for developing gene function hypotheses, reviewing gene lists, and ranking genes for functional experiments.

7) The sentence “The statistics for the aforementioned analyses were based on network analyses with every approach having a function F attached to it that gives each node v a numerical value (v)” is a little redundant and could use rephrasing.

Thank you. We have changed this line and moved a little up. Now, this reads as 

Each method has a function F attached to it that gives each node v a numerical value (v). If a node's score, or F(u), is higher than that of another node, or F(v), then we can say that node's ranking is higher than that of that other node.

8)  In the sentence “With ensuing cufflinks-cuffdiff pipeline, we obtained approximately 70 DEGs among which 65 were up-regulated and 5 were down-regulated with an inherent p-value heuristics <=0.05 and <=-2 Log2FC >=Log2FC” it might be useful to specify better the criteria used to identify DEGs. For example, it could be explained in more detail how the p-values and Log2FC were calculated and their biological meanings.

Thank you. The DEGs were sorted based on the p-value less than or equal to 0.05. The log2 FC were calculated based on this formula (used by cuffdiff command line)

log2(FPKMy/FPKMx) The (base 2) log of the fold change y/x 

9) Bibliographic data in the reference list needs a thorough review.

Thank you. We have checked all the references. 

Reviewer 5 Report

The manuscript written by Shukla et al. aims to identify and understand the critical signature pathways in Pca patients using DEGs analysis. While I found the topic/goal interesting, I have several concerns regarding the study results and conclusions. Please see my comments below for details:

1) Results 3.1 has a table with top-up-regulated DEGs. The column name for the 4th column is "p-value" is this raw p-value? Usually, for DEGs, authors are encouraged to report adjusted p-value considering the inflation of the false positive rate when conducting multiple comparisons. Without an adjusted p-value, it is tough to justify whether or not the change is significant. Therefore, please consider adding a column showing adjusted p values.

2) Assuming the supplementary data provides a complete table of all the DEGs obtained from this study and Table 2 under results 3.1 shows part of them (ranked as top-upregulated). Could the authors please explain why, instead of mentioning the genes listed in Table 2, which are considered "top," they chose to focus on genes such as "DOK5" or "STX6" later in the study that are not ranked top and not displayed in Table 2? If DOK5 and STX6 are closely linked with Pca, why is it not shown in Table 2? I suggest the authors revise this table so that all the genes discussed later in this manuscript are demonstrated.

3) In results 3.2, the authors picked some of the "prominent" genes identified via networks - DOK5, APP, CTNND1, STX6, STX10, STX16, BACE1, and BACE2. Could the authors please elaborate or define their criteria for "prominent"? Because I fail to see the rationale here. Judging by Figure 3, several genes were ranked at "1" by the network analysis. However, APP is not listed there, nor do STX16 and BACE2. Meanwhile, BACE1 was rated as "8" in Figure 3c. If their ranks do not define the "prominent," how should the readers interpret Figure 3? I suggest the authors add a paragraph of rationale to explain figure 3 and how they identify "prominent" genes using the information provided in figure 3 following results 3.2.

4) Figure 4 - why did the authors only pick those five genes out of many DEGs for such analysis? Could the authors please justify the reasons? 

5) Figure 6 - for each pie graph, could the authors please add a) the number of total DEGs analyzed and b) the percentage of genes each BP/MF category contains?

6) The section titled "Comparative analysis of RNA-seq data with other publicly available datasets" under results should be moved to the discussion. 

7) Supplementary PCR data - the color scheme needs to be revised to keep each bar graph consistent with the previous one. 

8) Supplementary DEGs table - please add fold change column and adjusted p-values. The reason for adding adjusted p-values is provided in my comment 1.

Author Response

Reviewer 5

The manuscript written by Shukla et al. aims to identify and understand the critical signature pathways in Pca patients using DEGs analysis. While I found the topic/goal interesting, I have several concerns regarding the study results and conclusions. Please see my comments below for details:

1) Results 3.1 has a table with top-up-regulated DEGs. The column name for the 4th column is "p-value" is this raw p-value? Usually, for DEGs, authors are encouraged to report adjusted p-value considering the inflation of the false positive rate when conducting multiple comparisons. Without an adjusted p-value, it is tough to justify whether or not the change is significant. Therefore, please consider adding a column showing adjusted p values.

Thank you. We have indeed used adjusted p-value. However those that were not bona fide with the p-value heuristics were removed.  

2) Assuming the supplementary data provides a complete table of all the DEGs obtained from this study and Table 2 under results 3.1 shows part of them (ranked as top-upregulated). Could the authors please explain why, instead of mentioning the genes listed in Table 2, which are considered "top," they chose to focus on genes such as "DOK5" or "STX6" later in the study that are not ranked top and not displayed in Table 2? If DOK5 and STX6 are closely linked with PCa, why is it not shown in Table 2? I suggest the authors revise this table so that all the genes discussed later in this manuscript are demonstrated.

Thank you for your suggestion. Now, we have added all the DEGs in table 2.

3) In results 3.2, the authors picked some of the "prominent" genes identified via networks - DOK5, APP, CTNND1, STX6, STX10, STX16, BACE1, and BACE2. Could the authors please elaborate or define their criteria for "prominent"? Because I fail to see the rationale here. Judging by Figure 3, several genes were ranked at "1" by the network analysis. However, APP is not listed there, nor do STX16 and BACE2. Meanwhile, BACE1 was rated as "8" in Figure 3c. If their ranks do not define the "prominent," how should the readers interpret Figure 3? I suggest the authors add a paragraph of rationale to explain figure 3 and how they identify "prominent" genes using the information provided in figure 3 following results 3.2.

The Cytohubba was used to check the top ranking genes. However by “prominent” we meant that they were either associated with PCa or cancer and here we found a few genes that could be crossed. We argue that these genes could be pleiotropic in nature. 

4) Figure 4 - why did the authors only pick those five genes out of many DEGs for such analysis? Could the authors please justify the reasons? 

Thank you. We have done the analysis for all our DEGs but due to space constraint, we could not put in the main text. We  have now attached them as a supplementary figure.

5) Figure 6 - for each pie graph, could the authors please add a) the number of total DEGs analyzed and b) the percentage of genes each BP/MF category contains?

Thank you. We have added this information in a supplementary file.

6) The section titled "Comparative analysis of RNA-seq data with other publicly available datasets" under results should be moved to the discussion. 

Thank you. We have moved this section to the discussion. 

7) Supplementary PCR data - the color scheme needs to be revised to keep each bar graph consistent with the previous one. 

Thank you. We have changed the control and malignant bars to different colors. 

8) Supplementary DEGs table - please add fold change column and adjusted p-values. The reason for adding adjusted p-values is provided in my comment 1.

Thank you. Now, we have added all the DEGs in main table 2.

Round 2

Reviewer 2 Report

The work of Shukla et al has improved on the minor concerns, but major concerns remain. Next, I kept the comment numbering of my previous review of this manuscript and noted the unresolved concerns. Given the authors were not able to address the major concerns, my recommendation is negative.

1.       This comment is properly addressed.

2.       Was there any correction for multiple hypotheses? If not, why?

Despite the authors’ answer, I do not see any q-value calculation in the manuscript. The authors should use a process, such as the Benjamini-Hochberg Procedure, to find significance amongst the Cufflink-Cuffdiff p-values. This is very important, because the distribution of p-values, especially for the samples 1 and 4, suggest that the p-value cutoff of 0.05 is not sufficient. Why samples 1 and 4 have so different p-value distributions from samples 2 and 3?

3.       Table 2 (now Table 3) shows part of the identified DEGs. What are the rest? Why only these are shown? How they were ordered?

The Table has been expanded. However, this expansion does not address any of the concerns. The authors say in the text “Some of the important DEGs are listed below”, but in their reply “Now, we have included all the genes in table”. To resolve this contradiction, I counted 58 up-regulated DEGs in contrast to 65 mentioned in the main text. So, why only some DEGs are shown here? Given that this data is available in supplementary tables, there is little point in repeating it in the main text. However, the authors could include selected DEG in the main text if there is a special interest about them and this was the spirit of my question. They could be genes with known associations to PCa or any cancer type, genes that form pathways of interest, or simply the top 5 DEGs in terms of significance and fold values. If so, these criteria should be clearly stated. Also, how the DEGs are ordered remains unclear. Also, what sample(s) these DEGs come from? The text says Table S3, but there are no numbers in the supplemental tables.  

4.       For Figure 2 (now figure 3), it is unclear what A, B, C, and D panels correspond to. Why these genes/interactions are shown and what is the conclusion of the analysis? Is there an interaction enrichment in DEG with high p-values compared to random p-values?

The authors answer did not address this comment. The purpose of having this figure and its interpretation remain unclear.

5.       Similarly, it is hard to follow what the hypothesis, test, and result is in the analysis shown in Figure 3. How they were defined and which are the “top network genes” and the “top 20 hub genes”?

The authors answer did not address this comment. The hypothesis, test, and conclusions for this figure remain unclear.

6.       What is the outcome of the analysis using TCGA dataset by cbioportal? How this helps the reader? I could say the same for the rest figures (Fig 4-8) of the manuscript.

The conclusion from this analysis is still unclear.

7.       This comment is properly addressed.

8.       This comment is properly addressed.

Author Response

Was there any correction for multiple hypotheses? If not, why? Despite the authors’ answer, I do not see any q-value calculation in the manuscript. The authors should use a process, such as the Benjamini-Hochberg Procedure, to find significance amongst the Cufflink-Cuffdiff p-values. This is very important, because the distribution of p-values, especially for the samples 1 and 4, suggest that the p-value cutoff of 0.05 is not sufficient. Why samples 1 and 4 have so different p-value distributions from samples 2 and 3?

There was multiple hypothesis done. The adjusted p value was considered as q value. However the q values that were not significant were not retained with P value Bonferroni correction. That is the reason why we found significant variation . Thank you 

  1. Table 2 shows part of the identified DEGs. What are the rest? Why only these are shown? How they were ordered? The Table has been expanded. However, this expansion does not address any of the concerns. The authors say in the text “Some of the important DEGs are listed below”, but in their reply “Now, we have included all the genes in table”. To resolve this contradiction, I counted 58 up-regulated DEGs in contrast to 65 mentioned in the main text. So, why only some DEGs are shown here?

We have removed this sentence “Some of the important DEGs are listed below” from the main text as we have mentioned all the genes now in table 3. Sorry for the confusion. Also, there were some DEGs which were left by mistake, now we have added and the list is complete.

Given that this data is available in supplementary tables, there is little point in repeating it in the main text. However, the authors could include selected DEG in the main text if there is a special interest about them and this was the spirit of my question. They could be genes with known associations to PCa or any cancer type, genes that form pathways of interest, or simply the top 5 DEGs in terms of significance and fold values. If so, these criteria should be clearly stated. Also, how the DEGs are ordered remains unclear.

Thank you for your suggestion. Initially, we filtered the DEGs based on p-value less than or equal to 0.05, which resulted in identification of 65 upregulated and 5 downregulated genes. Initially, in the manuscript, we had only highlighted the top enriched genes (around 10 genes) which are involved either in PCa or in some other cancer but based on some other reviewer’s suggestion, we included all the genes. 

 Also, what sample(s) these DEGs come from? The text says Table S3, but there are no numbers in the supplemental tables.  

Table S3 was the old table, now we had given all DEGs in a new excel sheet titled all DEGs.xls. We have also corrected this in the main manuscript as well. At the bottom of the excel sheet (all DEGs.xls), the sample number has been mentioned. It says sample 1, sample 2, sample 3 and sample 4. All these 70 DEGs come from all these different malignant samples (after annotation) which have been mentioned separately as well. Hope this is clear now. 

  1. For Figure 2, it is unclear what A, B, C, and D panels correspond to. Why these genes/interactions are shown and what is the conclusion of the analysis? Is there an interaction enrichment in DEG with high p-values compared to random p-values?

Thank you. The panels A, B, C and D correspond to four different malignant samples (A-69/19, B-1631/H19, C-4226/H19, D-5110/H20). GeneMANIA is a versatile, user-friendly web tool for developing gene function hypotheses, reviewing gene lists, and ranking genes for functional experiments. Through this, we identified several interacting partners which are also involved in different cancer pathways, which gave us a strong reason for further analysis of these genes. 

The authors answer did not address this comment. The purpose of having this figure and its interpretation remain unclear.

Thank you. We used GeneMania to identify any interacting partners of the DEGs. We get genes which are either co-expressed, physically interacting or co-localized. Co-expression means that two genes are linked if their expression levels are similar across conditions in a gene expression study. Similarly, physical interaction indicates that two gene products are linked if they were found to be interacting. For eg, COL6A1 interacts with COL6A2, COL5A1, KLF17, DNAJB11 etc. DNAJB11 is associated with a variety of signaling pathways, including aberrant signaling pathways associated with cancer. Through this interaction map, we identify new partners which might play important roles in different pathways and if we further want to choose these genes for functional validation in vitro.

  1. Similarly, it is hard to follow what the hypothesis, test, and result is in the analysis shown in Figure 3. How they were defined and which are the “top network genes” and the “top 20 hub genes”?

Thank you. When we queried all the DEGs across the samples, we chose the top 10 DEGs based on “Clustering coefficient method” through Cytohubba in Cystocape.  

The authors answer did not address this comment. The hypothesis, test, and conclusions for this figure remain unclear.

The cytoHubba Cytoscape plugin was used to better understand how all genes belonging to individual selected modules, that on the whole represent the macro-module, interact with each other. Moreover, cytoHubba was used to identify the most interconnected genes in the considered network, thus suggesting their biological involvement in the onset of traits significantly related to that particular disease. In this regard, cytoHubba allowed us to detect hub genes (i.e. genes that had the most connections to other genes of the network) and their subnetwork. 

  1. What is the outcome of the analysis using TCGA dataset by cbioportal? How this helps the reader? I could say the same for the rest figures (Fig 4-8) of the manuscript.

The TCGA is a gold standard dataset for retrieving and identifying causal genes in cancers. We were intrigued to check whether or not there are any DEGs in lieu of PCa pathogenesis in Cbioportal. Hence we have considered Cbioportal.  

The conclusion from this analysis is still unclear.

Thank you. We have now extended the discussions. Cbioportal is an open source for exploration of multidimensional cancer genomics data sets. We identified several genes from our study, we wanted to check what all kinds of mutations are present in those genes across a cohort of patients and for that cbioportal is a great platform where we have large data from different studies. For eg, through cbioportal we found out that COL6A1 has two missense (Variant of Unknown significance: VUS) mutations. We can also find out pathway alteration in cbioportal. This corroborates our findings and if we further want to use this information in our future studies. Similarly, Fig 5 also describes alteration frequency from cbioportal.

Fig 6 and 7 is GO analysis by Pantherdb which describes the role of DEGs in biological process and molecular function. We have also attached a supplementary table which shows the percentage of genes in different pathways. 

Reviewer 5 Report

Thank the authors for taking the time to address my comments. I have no further comments regarding this manuscript.

Author Response

Thank you very much. We are grateful for your kind suggestions!  

Round 3

Reviewer 2 Report

It is a disappointment to see no improvements in the manuscript, despite the authors’ efforts to reply to my comment. Therefore, my recommendation remains negative.

1.       Was there any correction for multiple hypotheses? If not, why?

I still see no q-value calculation or even mention in the manuscript, but the authors claimed: “There was multiple hypothesis done. The adjusted p value was considered as q value.” If they have done so, why the authors are hiding the “q-value” calculations? This is a critical issue and the authors should address it convincingly and in full detail in the manuscript.

Why samples 1 and 4 have so different p-value distributions from samples 2 and 3? The author’s reply did not address this point. The authors should also think how to address this issue in the manuscript.

 2.       Table 2 (now Table 3) shows the identified DEGs. How they were ordered?

This is the third time I am asking the same question: How they DEGs were ordered? They should be ordered in a meaningful way.

3.       For Figure 2 (now figure 3), it is unclear what A, B, C, and D panels correspond to. Why these genes/interactions are shown and what is the conclusion of the analysis? Is there an interaction enrichment in DEG with high p-values compared to random p-values?

How a reader should know that “The panels A, B, C and D correspond to four different malignant samples (A-69/19, B-1631/H19, C-4226/H19, D-5110/H20).”?

The authors answers regarding the purpose and interpretation of this figure are not reflected in the manuscript.

4.       Similarly, it is hard to follow what the hypothesis, test, and result is in the analysis shown in Figure 3. How they were defined and which are the “top network genes” and the “top 20 hub genes”?

The authors answer is still very abstract. The hypothesis, test, and conclusions for this figure remain unclear. The text is not updated.

5.       What is the outcome of the analysis using TCGA dataset by cbioportal? How this helps the reader? I could say the same for the rest figures (Fig 4-8) of the manuscript.

The conclusion from this analysis is still unclear. How the fact that COL6A1 has two VUS missense mutations in TCGA dataset corroborates the authors’ findings? What is the significance of this observation?

Author Response

Responses

It is a disappointment to see no improvements in the manuscript, despite the authors’ efforts to reply to my comment. Therefore, my recommendation remains negative.

Thank you very much for your comments. We have paraphrased the sections.  We have also changed the language and text for brevity besides keeping genes in italics as per HUGO criteria. 

  1.       Was there any correction for multiple hypotheses? If not, why?

I still see no q-value calculation or even mention in the manuscript, but the authors claimed: “There was multiple hypothesis done. The adjusted p value was considered as q value.” If they have done so, why the authors are hiding the “q-value” calculations? This is a critical issue and the authors should address it convincingly and in full detail in the manuscript. Why samples 1 and 4 have so different p-value distributions from samples 2 and 3? The author’s reply did not address this point. The authors should also think how to address this issue in the manuscript.

We apologize for this and realize the girth of the problem. We thought we had dealt with this statement earlier and have added it in the manuscript. We have included the whole set of individual files. We have paraphrased the section now and reason the following

While there are several approaches to check the multiple testing,  the false discovery rate of adjusted p-value is to reduce the p-value threshold.  The traditional approaches in our humble opinion, like Bonferroni correction, are  conservative reducing the false positives,  but they also reduce the number of true discoveries. So the FDR based adjusted p-value or q-value in our case was sought but no significant similarities across all four pairs were found. However, the distribution is more seen and we expect to expect, by chance, to get p-values < 0.05. Even otherwise, with significant changes we may be unsure if a p-value < 0.05 represents a statistical measure and so as p-value flattens, the calculated q-value is measured and this can be helpful for measuring the magnitude of numbers  with no meaning, for example x-fold threshold  determining biological significance. With these four samples, the p<0.05 were based on an ideal and the evidence seemed reasonable based on what we have done with clustering coefficient. 

  1.       Table 2 (now Table 3) shows the identified DEGs. How they were ordered?

This is the third time I am asking the same question: How they DEGs were ordered? They should be ordered in a meaningful way.

Thank you. We are sorry if you have missed this earlier. We have mentioned it in the methods as well.  First all columns with p value were sorted and the lowest p and adjusted p-value ( q-value) are indicative of the top enriched genes.  The values with + were considered as upregulated and the ones with - were considered and downregulated. However, +2 to -2 cutoff was rendered to check those DEGS that may be constitutively expressed between them instead. 

  1.       For Figure 2 (now figure 3), it is unclear what A, B, C, and D panels correspond to. Why these genes/interactions are shown and what is the conclusion of the analysis? Is there an interaction enrichment in DEG with high p-values compared to random p-values?

How a reader should know that “The panels A, B, C and D correspond to four different malignant samples (A-69/19, B-1631/H19, C-4226/H19, D-5110/H20).”?

Thank you. We have added a section for this and now the legend also mentions. 

Protein-protein interaction of different samples using GeneMANIA with purple edges representing co-expression, green -genetic interaction, red - physical interaction and blue representing co-localization. (a-d represents different sample pairs from which DEGs were inferred, viz. 69/19, 1631/H19, 4226/H19, 5110/H20 respectively and  the circles with edges are the input genes whereas solid circles are the interacting partners).

The authors answers regarding the purpose and interpretation of this figure are not reflected in the manuscript.

  1.       Similarly, it is hard to follow what the hypothesis, test, and result is in the analysis shown in Figure 3. How they were defined and which are the “top network genes” and the “top 20 hub genes”?

Thank you. We have paraphrased it wherein the central genes are the hub genes ( that are bigger nodes). However, to check the role of these genes, we employed clustering coefficient which is shown in the next figure. 

The authors answer is still very abstract. The hypothesis, test, and conclusions for this figure remain unclear. The text is not updated.

 Thank you. The hypothesis testing was for cufflinks whereas for PPI networks wherein we used GeneMania and employed correlation cutoff 0.95  

  1.       What is the outcome of the analysis using TCGA dataset by cbioportal? How this helps the reader? I could say the same for the rest figures (Fig 4-8) of the manuscript.

 Our immediate goal after the penultimate analyses of inferring DEGs was to ask whether any DEGs were relatively expressed in publicly available datasets from various studies.  Hence, we used TCGA wherein the PanCancer Atlas  dataset for Prostate Adenocarcinoma constituting 489 samples were screened for their functional roles and molecular aberrations 

The conclusion from this analysis is still unclear. How the fact that COL6A1 has two VUS missense mutations in TCGA dataset corroborates the authors’ findings? What is the significance of this observation?

 Thank you. As we have earlier shown the role of COL6A1  from our pilot studies, we sought to determine its role in other datasets. As the TCGA dataset was already checked, we felt it was a good way to corroborate our findings.   The section is paraphrased as below

While some of them the DEGs are specifically known to be associated with PCa, we also discovered a few novel lncRNAs which need further investigation. Expression of COL6A1 is significantly elevated in different tumors such as lung, prostate, cervical and pancreatic cancer compared to normal tissues. Interestingly, our previous WES studies in PCa have identified COL6A1 as one of the causal genes [7], whereas we also identified this through our RNA-seq analysis. COL6A1 was shown to be physically interacting with DNAJB11, APP along with some other genes. DNAJB11 is involved in aberrant signaling pathways associated with different cancers. Similarly, APP is known to be associated with androgen-responsive genes and regulates proliferation and migration of PCa cells [78].